# The Individual Effects of Cyclin-Dependent Kinase Inhibitors on Head and Neck Cancer Cells—A Systematic Analysis

**DOI:** 10.3390/cancers13102396

**Published:** 2021-05-15

**Authors:** Nina Schoenwaelder, Inken Salewski, Nadja Engel, Mareike Krause, Björn Schneider, Michael Müller, Christin Riess, Heiko Lemcke, Anna Skorska, Christina Grosse-Thie, Christian Junghanss, Claudia Maletzki

**Affiliations:** 1Department of Internal Medicine, Medical Clinic III—Hematology, Oncology, Palliative Medicine, University Medical Center Rostock, 18057 Rostock, Germany; Inken.salewski@med.uni-rostock.de (I.S.); mareike.krause@uni-rostock.de (M.K.); Christin.riess@med.uni-rostock.de (C.R.); christina.grosse-thie@med.uni-rostock.de (C.G.-T.); christian.junghanss@med.uni-rostock.de (C.J.); claudia.maletzki@med.uni-rostock.de (C.M.); 2Department of Oral and Maxillofacial Surgery, Facial Plastic Surgery, University Medical Center Rostock, 18057 Rostock, Germany; Nadja.Engel@med.uni-rostock.de; 3Institute of Pathology, University Medical Center Rostock, Strempelstr.14, 18057 Rostock, Germany; bjoern.schneider@med.uni-rostock.de; 4Core Facility for Cell Sorting & Cell Analysis, Laboratory for Clinical Immunology, University Medical Center Rostock, 18057 Rostock, Germany; Michael.Mueller2@med.uni-rostock.de; 5University Children’s Hospital, Rostock University Medical Centre, 18057 Rostock, Germany; 6Department of Cardiac Surgery, Reference and Translation Center for Cardiac Stem Cell Therapy (RTC), University Medical Center Rostock, 18057 Rostock, Germany; Heiko.Lemcke@med.uni-rostock.de (H.L.); anna.skorska@med.uni-rostock.de (A.S.); 7Department of Cardiology, University Medical Center Rostock, 18059 Rostock, Germany; 8Department Life, Light & Matter, Faculty of Interdisciplinary Research, University Rostock, 18059 Rostock, Germany

**Keywords:** targeted therapy, combination strategies, immunogenic cell death, xenograft model

## Abstract

**Simple Summary:**

This study examined the therapeutic potential of a combined therapy approach, based on clinical approved drugs (5-FU, Cisplatin, cetuximab) and cyclin-dependent kinase inhibitors (CDKi, dinaciclib, palbociclib, THZ1). We identified individual effects on head and neck squamous cell carcinoma cells, including induction of apoptosis/necrosis, and senescence as well as reduced invasiveness. Besides, we describe the relevance of the sequential timing of each combination partner to achieve synergistic effects. Another interesting finding of our study is the upregulation of immunologically relevant molecules on the tumor cell surface under certain CDKi-drug combinations. Here, dinaciclib and palboclicb had highest impact on immunogenicity, which even exceeded effects of the standard drugs. Finally, a therapeutic in vivo approach partially confirmed cell line-based results. Here, effective tumor growth control was seen when cisplatin was combined with dinaciclib. However, antitumoral effects were highly individual and nicely confirm the heterogeneity of this tumor entity.

**Abstract:**

Cyclin-dependent kinase inhibitors (CDKi´s) display cytotoxic activity against different malignancies, including head and neck squamous cell carcinomas (HNSCC). By coordinating the DNA damage response, these substances may be combined with cytostatics to enhance cytotoxicity. Here, we investigated the influence of different CDKi´s (palbociclib, dinaciclib, THZ1) on two HNSCC cell lines in monotherapy and combination therapy with clinically-approved drugs (5-FU, Cisplatin, cetuximab). Apoptosis/necrosis, cell cycle, invasiveness, senescence, radiation-induced γ-H2AX DNA double-strand breaks, and effects on the actin filament were studied. Furthermore, the potential to increase tumor immunogenicity was assessed by analyzing Calreticulin translocation and immune relevant surface markers. Finally, an in vivo mouse model was used to analyze the effect of dinaciclib and Cisplatin combination therapy. Dinaciclib, palbociclib, and THZ1 displayed anti-neoplastic activity after low-dose treatment, while the two latter substances slightly enhanced radiosensitivity. Dinaciclib decelerated wound healing, decreased invasiveness, and induced MHC-I, accompanied by high amounts of surface-bound Calreticulin. Numbers of early and late apoptotic cells increased initially (24 h), while necrosis dominated afterward. Antitumoral effects of the selective CDKi palbociclib were weaker, but combinations with 5-FU potentiated effects of the monotherapy. Additionally, CDKi and CDKi/chemotherapy combinations induced MHC I, indicative of enhanced immunogenicity. The in vivo studies revealed a cell line-specific response with best tumor growth control in the combination approach. Global acting CDKi’s should be further investigated as targeting agents for HNSCC, either individually or in combination with selected drugs. The ability of dinaciclib to increase the immunogenicity of tumor cells renders this substance a particularly interesting candidate for immune-based oncological treatment regimens.

## 1. Introduction

Mammalian cell cycle is controlled by cyclin dependent kinases (CDKs) [1]. In tumors, CDKs are dysregulated and CDK/cyclin complexes frequently overexpressed [2,3,4]. Tumor cells bypass the CDK4/6-Rb axis because it is critical for cell cycle entry and cell proliferation [5]. The knowledge about these mutations is a chance to identify molecular targets for pharmacological interventions [6]. Indeed, several CDK inhibitors (CDKi’s) been developed for cancer treatment. Additionally to the highly selective and FDA-approved CDKi’s palbociclib, ribociclib, and abemaciclib, multi- and pan-CDKi’s are now entering clinical trials. These include, among others, dinaciclib that targets CDK1, CDK2, CDK5, and CDK9 [7,8], and THZ1, which is active against CDK7, CDK12, and CDK13 [9,10].

Advances in understanding of pathobiology and molecular characteristics have contributed to the introduction of novel therapy approaches. Still, the treatment of solid tumors remains challenging. Additionally to intrinsic resistance mechanisms, the development or outgrowth of single subclones after therapy promotes immune escape and complicates precision medicine. 

Head and neck cancers are paradigmatic for tumor heterogeneity. They can be found in the oral cavity, pharynx, larynx, salivary glands, nasal cavity, and paranasal sinuses [11]. The predominant histological type of head and neck tumors is squamous cell carcinomas (HNSCC) [11]. HNSCC is the 7th most common cancer worldwide [11,12,13]. Risk factors include tobacco, alcohol, and human papillomavirus (HPV) infection. The latter drives tumor formation in the oropharynx with distinct clinical, histopathological, and molecular characteristics [14,15]. Around 58% of the patients present with loco-regionally advanced disease at diagnosis and this patient cohort has a poor prognosis [11]. Hence, the implementation of targeted therapies in standard treatment schedules constitutes a promising and urgently needed approach for improving treatment and outcome. In 2019, a multicenter, multigroup, phase 2 trial reported promising activity outcomes in patients with platinum-resistant or cetuximab-resistant HPV-unrelated HNSCC receiving palbociclib and cetuximab [16]. Though combination strategies are promising, the sequential timing of each combination partner remains debatable [17,18,19]. To move forward, we here employed simultaneous and sequential combination strategies of clinically approved therapeutics and CDKi’s for treating HNSCC with the aim to identify the best strategy. 

## 2. Results

### 2.1. CDKi Treatment Impairs Viability and Exerts Synergistic Effects in Combination Therapy

UT-SCC-14 and UT-SCC-15 were used as in vitro cell culture models, since these cells are representative for primary and recurrent HNSCC. Both cell lines were susceptible to standard drugs and CDKi’s in clinically relevant doses (below 1 µM for CDKi’s and ≤90 µg/mL for cytostatic drug), as determined in preliminary experiments. For combination experiments, standard drugs 5-Fluorouracil (5-FU), Cisplatin, and cetuximab as well as CDKi’s (dinaciclib, palbociclib, THZ1) were applied in doses below the IC_50_ (Figure 1A,B; cetuximab is the only exceptions, here IC_50_ doses were used). The time course of treatment considered each cell lines’ doubling times and attempted to mimic the in vivo situation. Therefore, cells received two treatment cycles of 72 h.

In a first series, simultaneous combinations were applied (Figure 1B). Notably, dual CDKi treatment was synergistic or additive in UT-SCC-14 and partially in UT-SCC-15 cells as determined by biomass quantification. Here, combinations of dinaciclib with palbociclib or THZ1 were synergistic (Figure 1B). CDKi/drug combinations were mainly antagonistic. The only exception was seen for Cisplatin in conjunction with dinaciclib (UT-SCC-14) and cetuximab with dinaciclib or THZ1 (UT-SCC-15). 

To test if the effect of a 2 × 72 h CDKi monotherapy can be boosted, sequential combinations were performed (Figure 1C–E). The first bar of each graph shows CDKi monotherapy, followed by the sequential combination treatments. CDKi were either given before or after standard therapy. The sequential treatment with dinaciclib (Figure 1C) revealed higher biomass reduction in both cell lines when standard therapy was given first. There was a strong reduction for all three combinations in UT-SCC-14 and UT-SCC-15. The sequential treatment with palbociclib yielded opposite results (Figure 1D). Here, palbociclib pretreatment prior to Cisplatin or cetuximab was better than the other way around. The order of 5-FU application had no leverage. For the sequential combination with THZ1 (Figure 1E), cell line-specific responses were seen. UT-SCC-14 cells’ viability was more affected when 5-FU was given first and second THZ1. Comparable effects were seen after THZ1/Cisplatin treatment in UT-SCC-15 cells. Still, the other combinations were only effective when the standard drug was given before. 

The aforementioned findings nicely confirm the heterogeneous response pattern of HNSCC.

### 2.2. CDKi’s Induce Apoptotic and Necrotic Cell Death and Mediate Calreticulin Translocation

To investigate the effects of different treatments on the two cell lines, an apoptosis-necrosis assay was performed on selected treatment schedules (Figure 2A,B). Cells were simultaneously treated with CDKi’s and drugs (5-FU, Cisplatin) for 24 and 72 h (Figure 2A,B). Short-term dinaciclib monotherapy mainly induced early apoptotic and necrotic cell death. The other monotherapies had minor or no impact on cell viability. After 72 h, overall cell death was higher in treated cells, but with individual differences. Dinaciclib alone or in combination induced necrosis, THZ1 and its combinations triggered apoptosis or a mixed form of apoptosis and necrosis (Figure 2A,B). Additionally to the induced cell death, senescence was studied, since this is a common response to CDK inhibition (Appendix A). These experiments revealed senescence induction by specific CDKi’s (e.g., dinaciclib) or its combination with standard drugs (e.g., 5-FU). However, senescence was not the dominating cellular response here, suggesting a minor role. UT-SCC-14 cells clumped together, especially under dinaciclib monotherapy and combination therapy, while UT-SCC-15 cell clusters were disrupted. The combination of THZ1 and 5-FU had similar effects to dinaciclib. 

Then, the ability to induce immunogenic cell death was measured after 72 h by detecting calreticulin (CalR) on the tumor cells’ surface (Figure 2C). The proportion of CalR positive cells and the mean fluorescence intensity signal (MFI) (Figure 2D) were recorded. Dinaciclib induced CalR translocation in monotherapy and combination therapy significantly. Notably, the combination of THZ1 and 5-FU likewise induced CalR translocation. While these findings already hint towards immune stimulating properties, we additionally checked for immunologically relevant markers (Figure 3A,B). The abundance of HLA-ABC (MHC class I) and PD-1 on tumor cells was examined. A significant increase in MHC class I was seen after dinaciclib monotherapy and combination therapy as well as upon palbociclib treatment of UT-SCC-14 cells (Figure 3A). The MHC class I abundance changed marginally in UT-SCC-15 cells irrespective of the treatment schedule used (Figure 3B). This was, however, likely because of the high basal MHC class I abundance, which was about 80%. Still, dinaciclib and their combinations tended to upregulate MHC class I, finally yielding ~100%. PD-1 was upregulated by certain treatments. This did, however, not reach statistical significance (Figure 3A,B). 

### 2.3. CDKi Induce Cell Cyclce Arrest

Due to the mode of action of CDKi’s, cell cycle analysis was done on residual tumor cells (typically below 50%; Figure 3C–E). Representative histograms for all treatments are given in Appendix A. The number of residual cells after dinaciclib treatment was low. In these, tumor cells’ cycle distribution was quite similar to controls. In UT-SCC-14 cells, a lucid G1 arrest was only seen after combined THZ1 and 5-FU therapy (*p* < 0.05 vs. control), while the remaining treatments had a minor impact on the cell cycle. UT-SCC-15 cells had significant changes after palbociclib and 5-FU monotherapy, but not in the combinations. 

### 2.4. CDKi’s Have Minor Impact on Double-Strand Breaks and Radiosensitivity 

Treatment-induced double-strand breaks (DSBs) were determined by fluorescence microscopy using γ-H2AX (Figure 4). H2AX is phosphorylated by kinases after DNA double-strand breaks on serine 139. CDKi monotherapy or combination therapy itself had minor impacts on γ-H2AX foci, which were hardly detectable (Figure 4). To test if the applied regimens may enhance radiosensitivity, we then checked for irradiation-induced DSBs using 2 Gy (Figure 4B). Indeed, numbers of γ-H2AX-positive cells increased, with highest amounts in cells treated with palbociclib. With regard to the combinations, γ-H2AX foci were primarily seen in palbociclib-or THZ1- based combinations with 5-FU. By contrast, such radiosensitizing effects were not seen in combinations with dinaciclib and may thus constitute a specific consequence of palbociclib or THZ1 treatment. 

### 2.5. CDKi’s Remodel the Actin Filament

Live cell monitoring via impedance measurements is particularly suitable for studying alterations in the cell monolayer, in the adhesion properties, and in the membrane integrity in real time. While the impedance increased over time in untreated control cells, dinaciclib treatment massively reduced impedance (Figure 5A,B). For palbociclib treated UT-SCC-14 cells, the measured impedance slightly decreased after 48 h, while THZ1 monotherapy slightly increased impedance (Figure 5A). Notably, the combination of THZ1 and 5-FU caused a delayed impedance breakdown in both cell lines. Here, impedance increased within the first 20 h. Thereafter, the impedance stagnated for approximately 3 h, and then decreased for the next 48 h until no impedance was detectable. Dinaciclib in conjunction with cytostatics (Cisplatin, 5-FU) induced a complete and irreversible breakdown. 

To confirm the impedance data, actin fibers were stained 72 h after treatment (Figure 5C,D). Untreated UT-SCC-14 cells form a typical monolayer with a cortically formed cytoskeleton and less stress fibers within the cells. Dinaciclib itself caused massive cell detachment and consequently cell death. Nearly all UT-SCC-14 cells were detached after dinaciclib treatment, while some UT-SCC-15 cells remained attached and spread. Cytostatics (Cisplatin, 5-FU) intensified actin abundance in both cell lines. 

THZ1 strengthened the formation of stress fibers in both cell lines that increases cellular stiffness and changes the motility properties [20]. This finding adds to the higher impedance under THZ1 treatment compared to the untreated control. THZ1 in combination with 5-FU caused higher cytotoxic effects, so most cells were detached. 

### 2.6. Influence on Mitochondria, Lysosomes, the Endoplasmatic Reticulum and Vacuole Formation 

CDKi-based treatments induced cytoplasmic vacuole formation. Hence, we checked the influence of the treatment schedules on mitochondria, lysosomes, and endoplasmatic reticulum (ER) (Figure 6). In both cell lines, mitochondrial activity increased after dinaciclib monotherapy and combination therapy (Figure 6). Monotherapy with palbociclib, THZ1, or Cisplatin induced lysosome formation, but only in UT-SCC-14 cells. An effect of the treatments on the ER could not be demonstrated. After 5-FU treatment, the mitochondrial activity of UT-SCC-15 cells slightly increased that was reversed by THZ1. Cisplatin monotherapy had opposite effects that were neutralized by the combination partners (Figure 6B).

Cells were stained for specific late endo-lysosomal markers CD107a (LAMP1) and Rab7a to confirm above findings (Figure 6C,D). The GTPase Rab7a is primarily associated with late endosomes and LAMP1 is typically considered lysosomal [21]. Cell line specific responses were observed, with UT-SCC-14 cells showing higher numbers of these late endo-lysosomal markers after treatment. In detail, dinaciclib, palbociclib, and 5-FU monotherapy resulted in the highest increase of positive cells (*p* < 0.05 5-FU vs. control) (Figure 6C). The combinations could not boost effects. In UT-SCC-15 cells, highest numbers of CD107a^+^/Rab7a^+^ cells were detected after dual CDK inhibition (palbociclib + dinaciclib) (Figure 6D), implying that lysosomal formation plays a minor role here.

#### 2.6.1. CDKi’s Reduce Invasiveness and Migratory Potential 

An assay was performed to explore the migration potential of cells to a cell free space under treatment. The cell line UT-SCC-14 filled the scratch within 24 h; the same was true for THZ1 and 5-FU monotherapy and combination therapy (Appendix A). The toxic activity of dinaciclib induced cell death within 72 h and an accordingly incomplete scratch closure. Adding THZ1 to dinaciclib delayed migration, so the scratch was filled after 48 h. Using an invasion assay, the ability of cells to escape from the toxic environment was then investigated. For this experiment, selective treatments were included based on the obtained results shown before. The invasive cells from treatment medium were put in relation to invasive cells from control medium. CDKi treatment with dinaciclib significantly reduced invasiveness (Appendix A). Effects were even stronger when two CDKi’s were combined (dinaciclib + THZ1), but not by adding 5-FU. Still, these data confirm the potential of CDKi’s to interfere with cellular invasion.

#### 2.6.2. In Vivo Results

Finally, a xenograft mouse model was used to test if in vitro results can be transferred in vivo. For this proof-of-concept study, dinaciclib and Cisplatin were chosen as therapeutics and given alone or in combination (Figure 7A). We decided to use this combination, since dinaciclib had strong antitumoral effects in all previous analyses and Cisplatin is the accepted standard of care for HNSCC patients.

UT-SCC-14 xenografts showed a poor treatment response. Monotherapy had no influence on tumor growth and the combination was only able to decelerate growth (Figure 7B). In contrast, UT-SCC-15 xenograft growth was significantly reduced under therapy (Figure 7B). Dinaciclib and its combination with Cisplatin decreased the tumor volume to a minimum, the latter even stopped tumor growth until the experimental endpoint (two months follow-up). As a consequence of the better treatment response, mice challenged with UT-SCC-15 lived longer compared to those harboring UT-SCC-14 xenografts (Figure 7C). The outcome was best in the combination with a median survival of 63 days (vs. control 42 days, *p* < 0.05). As for dinaciclib monotherapy, mice had to be euthanized mostly because of tumor ulcerations. Hence, the poorer survival in both cases is not justified by the tumor volume as an endpoint but due to ethical aspects. Histology of residual tumors confirmed the different treatment responses. UT-SCC-14 xenografts presented with initial necrosis that increased after dinaciclib treatment (Figure 7D). After Cisplatin therapy, beginning necrosis with initial inflammatory reaction was visible, but also vital tumor tissue. In the combination, keratinized squamous cell carcinoma containing degenerated cells was found. In addition, neutrophilic infiltration was observed. The UT-SCC-15 xenograft sections of control mice showed characteristics of a keratinizing squamous cell carcinoma with developing necrosis. After dinaciclib treatment, degenerated and early apoptotic cells became prominent with surrounding necrosis. Cisplatin monotherapy primarily induced necrosis. The dark spot in the center of the image is degenerated keratinized squamous epithelium. Necrosis was dominating in the combination with some swollen cells, indicative of early cell damage in the initial stage. 

## 3. Discussion

CDKi’s are being applied in clinical trials to treat solid and hematological malignancies (e.g., *NCT04169074*, *NCT04391595*, *NCT03981614*, and *NCT01627054*). For locally advanced or metastatic breast cancer, the CDK4/6 inhibitors palbociclib, ribociclib, and abemaciclib are FDA approved in combination with endocrine therapy [22,23]. Though approval for HNSCC is still pending, first preclinical and clinical reports are promising [16,24,25]. 

Our study adds another piece of evidence and identifies the CDK4/6 inhibitor palbociclib as well as the global acting CDKi’s dinaciclib and THZ1 as promising candidates for HNSCC treatment. We additionally describe the strong dependency on (a) the combination partner and (b) the temporal order of applying each substance to reach therapeutic effects. Notably, simultaneous dual CDKi treatment, but not the combination with standard drugs, worked synergistically in our settings. The sequential application yielded heterogeneous results, depending on the CDKi used for combination. In theory, chemotherapeutic drugs should benefit from prior CDKi treatment by completing their effects [26,27]. However, this was only seen here when the CDK4/6 inhibitor palbociclib was used as first treatment, confirming findings from a recent study in which intrinsic resistance was reported when Cisplatin was given before palbociclib because of drug-induced *c-Myc* and *Cyclin E* upregulation [19]. Though not analyzed in detail here, comparable molecular alterations can be anticipated. Besides direct antitumoral effects, another argument for applying specific CDKi’s in the first-line is the protection of normal hematopoietic stem and progenitor cells via transient G1 cell cycle arrest induction and the maintenance of antitumor immunity to boost the patient’s tolerability towards chemotherapy [28]. This “positive” side effect was recently observed in phase II trials on patients with small-cell lung cancer receiving the CDK4/6 inhibitor trilaciclib [28,29]. Hence, a favorable outcome can indeed be speculated if CDK4/6 inhibitors are applied in the first-line. For the more global acting CDKi’s, such beneficial responses are very unlikely. Instead, leukopenia and neutropenia were reported as direct consequences of the complex mode of action, including interference with RNA polymerase II binding [30,31,32]. These systemic toxicities constitute a major limitation and clinicians will have to cope with this challenge. With regard to the sequential application applied here, first-line chemotherapy was superior to second-line chemotherapy. This regimen was comparable to or even better than two cycles of dinaciclib or THZ1 monotherapy. Mechanistically, effects were due to early apoptosis with a shift to necrosis afterward. Quite in line with this, Hossain et al. also observed apoptosis induction by short-term dinaciclib treatment [33]. Notably, THZ1, had a minor impact on apoptosis, though this was described in literature in an nM range and thus was comparable to doses used here [34]. This might be best explained by some kind of delayed apoptosis induction but not intrinsic apoptosis resistance of our HNSCC cells. In support of this, the specific CDK4/6 inhibitor palbociclib triggered apoptosis in both cell lines, confirming recent observations [35]. Senescence, another CDKi-induced cellular stress response, was also seen here; however, it was not as profound as described in the literature [36]. Hence, senescence may either play a minor role in HNSCC or it was an early event and thus undetectable after two rounds of treatment. The strong cytotoxicity of dinaciclib and certain CDKi/drug combinations argue in favor of the latter. 

Quite in line with this, impedance reduced massively under dinaciclib monotherapy and combination therapy that was accompanied by remarkable changes in cell shape and cytoskeletal organization. This, in turn, may impair cell-cell contacts via adhesion molecules, electrical coupling, and passage through gap junctions [37]. A comparable, but delayed impedance breakdown was achieved when THZ1 was combined with 5-FU, likely because of the 5-FU’s mechanism of action [38]. Such a delayed effect under 5-FU treatment was also seen in the wound healing assay. Conversely, THZ1 monotherapy slightly increased impedance, accompanied by re-organization of cortical actin into stress fibers. These stress fibers increase the cellular stiffness and reduce the motility [20,39,40]. We therefore propose the identified shift in actin organization as main response towards drugs applied in this study that has to be addressed in more detail prospectively. By performing a direct comparison of the two cell culture models used here, it is obvious that the cell line UT-SCC-15, established from a nodal recurrence of a primary tongue carcinoma [41], shows more cortical actin than intracellular stress fibers. In the UT-SCC-14 cells, it is exactly the other way around. The cortical actin filaments are important to create tension, leading to gradients that generate changes in the shape which are important during cell migration, cell division, and tissue morphogenesis [42]. Also, we hypothesize that the remodeling of the actin filament makes the cells more vulnerable to immune cells. A prerequisite—among others—for this is the induction of immunogenic cell death (ICD) in tumor cells to activate phagocytes [33,43,44]. Actually, we observed increased Calreticulin (CalR) translocation upon combined THZ1/5-FU treatment. While this effect was not visible under monotherapy, we suggest this treatment regimen as a promising strategy for immunotherapeutic approaches. Notably, dinaciclib was similarly able to induce CalR translocation and upregulation of the immunologically relevant marker MHC class I to an extent comparable to THZ1/5-FU combination therapy. This makes dinaciclib particularly interesting in the context of immunotherapy, as hypothesized before [33,45]. Hossain et al. treated murine CT26 colon cancer cells for 24 h with different dinaciclib concentrations (0.05 µM–25 µM) and identified a linear increase in CalR translocation. The readout in this study was the mean fluorescence intensity, which was around 1200 after treatment with 0.05 µM dinaciclib [33]. In our work, a concentration of 0.02 µM dinaciclib yielded an MFI around 2000 for UT-SCC-14 and an MFI of around 1500 for UT-SCC-15 cells. Though the MFI is not directly comparable among different studies, it still confirms previous findings. Additionally to this, the observed upregulation of MHC class I enhances antigen presentation and ultimately stimulates CD8^+^-T-cells to finally promote antitumor immunity [46]. 

Radiotherapy is the mainstay of therapy for HNSCC patients and can be combined with immunotherapy. While radiotherapy itself has the potential to reprogram the tumor microenvironment, several drugs including CDKi’s have been identified as radiosensitizers [24,47,48]. However, the CDKi´s used here had a minor impact on double-strand breaks and radiosensitivity. Radiosensitization, if any, was seen after combining palbociclib or THZ1 with 5-FU. Wang et al. reported palbociclib-induced DNA damage in an p53-independent manner and repressed DNA damage repair ability via RAD51 downregulation [35]. THZ1 inhibits CDK7, CDK12, and CDK13 [49] and was described as radiosensitizer in a study on medulloblastomas [50]. Genes involved in homologous recombination such as Brca2, Rad51, and Rad50 were downregulated, accompanied by increased γH2AX-foci post irradiation [50]. Another study likewise confirmed increased amounts of γ-H2AX foci upon THZ1 treatment [51]. Hence, THZ1 has the potential to sensitize to radiation and impair recovery from radiation-induced DNA damage. The fact that another target of THZ1, CDK12, selectively controls the expression of genes involved in the DNA damage response, supports this theory [52]. The question remains why such effects were undetectable in our study. Apart from the radiation dose (2Gy), the time is a critical factor for detecting or missing a clear radiation response. Hence, it is conceivable that we have simply missed certain effects. 

Another common response towards CDKi treatment is cell cycle alteration. Palbociclib induced a G1 phase arrest that complies with its mode of action and has been described in the literature [53,54,55]. Combined THZ1 and 5-FU therapy yielded comparable results in both cell lines that can be explained as follows: 5-FU limits the availability of thymidylate and inhibits the DNA synthesis [56,57]. THZ1 impairs CDK2 activity via inhibition of CDK7 [58,59]. CDK2 is required for the transition from G1 to S phase, blocking this CDK thus holds the cell cycle [49]. This has profound biological effects. In a very recent study on patient-derived glioblastoma models, we described the CDKi-induced loss of mitochondrial function in pioneering work, characterized by a multivacuolar phenotype and signs of early-methuosis [60]. Methuosis, a non-apoptotic cell death phenotype, is defined by the accumulation of large fluid-filled cytoplasmic vacuoles that originate from macropinosomes [61]. With regard to the HNSCC cells used here, dinaciclib monotherapy and combination therapy strengthened the mitochondrial activity. However, methuosis did not seem to play a major role, as late endosomes and vacuoles markers CD107a and Rab7a exclusively increased under CDKi or 5-FU monotherapy. Hence, CDKi’s have indeed different effects on individual tumor cells. 

In a final in vivo proof-of-concept experiment, dinaciclib and Cisplatin were chosen based on the following criteria: Dinaciclib has complex effects on HNSCC tumor cells, including growth inhibition, prevention of migration/invasion, and cytotoxicity. Besides, dinaciclib is a potent ICD inducer and a promising candidate for combined immunotherapies. Cisplatin is a well-known cytostatic drug approved as 1st line HNSCC treatment and widely applied in the clinic [11,62,63]. Additionally, preclinical reports on combined dinaciclib-Cisplatin application are promising, as recently shown for a subcutaneous ovarian cancer xenograft model in nude mice [64]. Here, the combination approach was likewise superior to either single treatment and most effective in suppressing UT-SCC-15 growth. While this cell line was established from a nodal recurrence, it is tempting to speculate that advanced tumors may even benefit more from this regimen than lower-stage tumors. However, this has to be tested on a larger series of (matched) tumor samples. However, the accelerated growth of UT-SCC-15 xenografts under Cisplatin monotherapy is worth mentioning. Intrinsic resistance is unlikely, since these cells showed good sensitivity in vitro. Also, acquired resistance can be excluded because tumors grew constantly under treatment. Comparable effects were not reported in the literature. We can therefore only assume that outcomes may be improved by changing the treatment schedule (i.e., dose, application route, and the number of injections). Still, the complex interaction of tumor, normal stromal, and immune cells influences outcomes—a major contributing factor that can only partially be considered in vitro [65]. 

To sum up these findings, we provide another piece of evidence for the therapeutic activity of CDKi’s, their complex mode of action, and the rationale to combine targeted agents with “conventional” drugs or even immune-restoring approaches to succeed in the long run. 

## 4. Materials and Methods 

### 4.1. Tumor Cell Lines and Culture Conditions

Two patient-derived cell lines: UT-SCC-14 and UT-SCC-15, were used. The UT-SCC-14 was established from a primary tumor of the tongue and the UT-SCC-15 derived from a nodal recurrence of the same origin. Both cell lines are HPV negative. Cells were maintained in full medium: DMEM/HamsF12 supplemented with 10% fetal calf serum (FCS), glutamine (2 mmol/L) and antibiotics (medium and antibiotics were purchased from Pan Biotech, Aidenbach, Germany, FCS from Sigma-Aldrich, Darmstadt, Germany and glutamine from Biochrom, Berlin, Germany) and kept in low passages.

### 4.2. Cytostatic Drugs and Targeted Substances

The approved cytostatic drugs 5-FU (50 mg/mL) and Cisplatin (1 mg/mL), the approved therapeutic antibody cetuximab (5 mg/mL) and the targeted substances dinaciclib, palbociclib, and THZ1 (all 10 mM) were used. 5-FU, Cisplatin and cetuximab were purchased from the pharmacy of the University Hospital Rostock, dinaciclib and palbociclib from Selleckchem, Munich, Germany, and THZ1 from Hycultec, Beutelsbach, Germany. 

### 4.3. Dose Response Curves and Combination Therapy 

For dose response curves, cells were seeded in 96 well plates in three technical replicates per cell line and incubated for 24 h at 37 °C and 5% CO_2_. Afterwards, cells were treated for 2 × 72 h in monotherapy with the different test substances in concentrations ranging between 0.05 µg/mL and 1 mg/mL for approved drugs and 1 nM and 1 µM for CDKi’s. Thereafter, various combinations were tested in simultaneous and sequential settings. Doses used for combinations were as follows: 5-FU [0.32 µg/mL or 90 µg/mL], Cisplatin [0.05 µg/mL, 0.5 µg/mL, or 0.1 µg/mL], cetuximab [0.5 µg/mL], dinaciclib [0.005 µM or 0.02 µM], palbociclib [1 µM], and THZ1 [0.02 µM or 0.005 µM] depending on the treatment duration of each substance (1 × 72 h or 2 × 72 h). Readout was done by crystal violet staining. In sequential combination therapy, two different approaches were applied. Firstly, the cells were treated with the standard therapy for 72 h and the CDKi’s afterwards, and secondly, the administration was done in reverse order. To rule out the possibility that the single 72-h administration of the approved therapeutics is responsible for the potentially stronger effect, they were tested in monotherapy for 72 h. Potential synergistic or additive effects between the substances in a 2 × 72 h simultaneous combination approach were analyzed with the Bliss Independence model. 

### 4.4. γ-H2AX Staining

Tumor cells were treated for 24 h in Chamber Slides with selected concentrations and combinations of the test substances and then irradiated with 2 Gy (Cs-137 γ-irradiation; IBL 637, CIS Bio-International, Codolet, France). γ-H2AX staining was performed 6 h after irradiation. Cells were washed with phosphate-buffered saline (PBS), fixed in 4% paraformaldehyde w/o methanol (Thermo Scientific, Darmstadt, Germany) for 30 min, washed again, followed by cell permeabilization in 0.5% Triton X−100 (Sigma-Aldrich, Darmstadt, Germany) for 15 min. After blocking the unspecific binding sites with 1% bovine serum albumin (Serva, Heidelberg, Germany), cells were incubated with the monoclonal γ-H2AX antibody (1:100; BioLegend, San Diego, CA, USA) over night at 4 °C. Cells were washed and nuclei stained with 4′,6-diamidino-2-phenylindole (DAPI) (AAT Bioquest, Sunnyvale, CA, USA). Analysis was performed with a ZEISS Elyra 7 Confocal Laser Microscope (Zeiss, Jena, Germany).

### 4.5. Apoptosis-Necrosis Assay, Phenotyping, and Immunogenic Cell Death

Apoptosis-necrosis was determined after 24 and 72 h treatment, phenotyping was done after 48 h, and determination of immunogenic cell death (ICD) was recorded after 72 h treatment. Cells were analyzed on a Flow Cytometer (BD FACSVerse™, BD Pharmingen, San Jose, CA, USA). Data analysis was done using the BD FACSuite software (BD Pharmingen).

For Apoptosis-necrosis, cells were stained for 20 min at room temperature with 0.2 µM Yo-Pro 1 iodide (Thermo Scientific, Ex/Em 491/509 nm; blue laser 488 nm) and 20 µg/mL Propidiumiodide (PI) (Sigma-Aldrich, Darmstadt, Germany; Ex/Em: 535⁄ 617 nm; blue laser 488 nm). PI was added shortly before flow cytometry. For phenotyping, cells were stained for 30 min at 4 °C with FITC anti-HLA-ABC antibody (MHC I) (1:50; ImmunoTools, Friesoythe, Germany) and APC anti-CD279 (PD-1) (1:50; both from BioLegend, blue (488 nm) and red (633 nm) laser). ICD was detected by staining translocated CalR on the cell surface. Cells were incubated for 30 min at 4 °C with the polyclonal rabbit CalR primary antibody (1:50; Abgent, San Diego, CA, USA). Cells were washed and labeled with FITC-conjugated secondary antibody (donkey anti rabbit, 1:50; BioLegend), and incubated again for 30 min at 4 °C. In order to exclude non-specific binding of the FITC-labeled secondary antibody, control cells were additionally stained with the secondary antibody without using the primary antibody. For CalR quantification, the number of cells that were positive for the secondary antibody was subtracted from the CalR+ secondary antibody stained cells.

### 4.6. Cell Cycle Assay 

Cell cycle was determined after 48 h of treatment. Cells were harvested, counted, and resuspended with 1 mL ice cold 70% ethanol. Cells were incubated overnight at −20 °C, washed again, and incubated with 0.5 mL 0.25% TritonX-100 for 15 min on ice. Cells were washed and resuspended in RNAse A (100 μg/mL), supplemented with PI (20 μg/mL). After 30 min incubation on ice, cells were analyzed on a Flow Cytometer (FACSCalibur, BD, San Jose, CA, USA). Data analysis was done using BD FlowJo software (BD Pharmingen, San Diego, CA, USA). 

### 4.7. Influence on Mitochondria, Lysosomes, ER, and Vacuole Formation

The influence on mitochondria, lysosomes, and the ER was examined with immunofluorescence staining. Cells were seeded in Chamber Slides and stained after 72 h treatment. Then, cells were washed and the staining with MitoTracker Red CMXRos (20 nM, CellSignaling Technology, Danvers, MA, USA) and ER-Tracker Blue-White DPX (1 µM, Invitrogen) was done simultaneously for 35 min at 37 °C. Cells were washed and stained with LysoTracker DND-26 (50 nM, CellSignaling Technology) for 2 min at room temperature. Analysis was performed on a ZEISS Elyra 7 Confocal Laser Microscope (Zeiss).

Additionally, vacuole formation was analyzed after 72 h treatment using specific antibodies. Cells were harvested and incubated with Alexa488 anti-CD107a antibody (Biolegend, 1:50 in 0.1% BSA) for 30 min at 4 °C. Then, cells were washed and resuspended in 0.5 mL FluorFix^TM^ Buffer (Biolegend) for 20 min at room temperature. Afterwards, cells were washed twice with 1× intracellular staining perm wash buffer and incubated with Alexa594 anti-Rab7a antibody (Biolegend, 1:50 in 0.1% BSA) for 30 min at room temperature. The reaction was stopped with PBS and washed before cells were resuspended in 200 μL PBS (+2 mM EDTA). Cells were analyzed by flow cytometry on a Flow Cytometer (FACSAriaII, BD, blue (488nm) and yellow-green (561 nm) laser). Data analysis was performed using BD FACSDiva software (BD).

### 4.8. Senescence

Senescence-associated β-galactosidase (SA-β-gal, Cell Signaling Technology, Cambridge, UK) was analyzed after 72 h of treatment. Cells were washed and fixed. After a second washing step, cells were stained with a Galactosidase Staining Solution. Therefore, cells were incubated at 37 °C overnight in a dry incubator and checked for senescence the following day under a microscope. To analyze the number of senescent cells, ImageJ was used. 

### 4.9. Impedance Measurement and Actin Staining

Impedance was measured with a commercial Electric Cell-Substrate Impedance Sensing system (ECIS Zθ; Applied Biophysics, New York, NY, USA) equipped with a 96-well array station (Applied Biophysics) to monitor time and frequency dependent complex impedance, Z (t, f). Cells were grown on a 96-well ECIS array plate with 20 interdigitated electrodes/well (96W20idf PET; ibidi GmbH, Gräfelfing, Germany). Prior to cell seeding, electrodes were stabilized with serum-free media overnight in the incubator with high humidity at 37 °C and 5% CO_2_. Impedance measurements were performed directly in the treatment medium in the incubator, allowing real-time monitoring of all impedance alterations at 11 frequencies (0.0625, 0.125, 0.25, 0.5, 1, 2, 4, 8, 16, 32, and 64 kHz) in a 180-s interval. 24 h after cell seeding, treatment was added for 72 h. Analysis of cell-cell contacts was performed by 4000 kHz using ECIS Software (Applied Biophysics). 

To confirm the results of the impedance measurement, the actin filament was stained with phalloidin (1:300; Invitrogen, Darmstadt, Germany). Therefore, cells were treated for 72 h in Chamber Slides, fixed, permeabilized, stained, and analyzed as described for γ-H2AX. 

### 4.10. Wound Healing and Invasion Assay

A wound healing assay was done in 12-well plates. After formation of a confluent cell layer, a defined scratch was set. Medium was removed, cells were washed with cell culture media, and the corresponding treatment based on the most promising simultaneous combinations was added. Scratch closure was documented by light microscopy routinely during the following 72 h.

For the invasion assay, inserts (8.0 µm translucent; Greiner bio-one, Frickenhausen, Germany) were coated with 70 µL Matrigel (1:25 in serum free media; Corning, NY, USA) and cells seeded in serum free, treatment containing media. The inserts were placed in a 24-well plate containing 750 µL media with 10% FCS and incubated for 72 h. Invasiveness was analyzed by WST-1 assay. The inserts were placed into a new 24-well plate containing WST-1 in serum free media. WST-1 containing medium without cells served as a blank. After 2.5 h of incubation, absorption was measured at a wavelength of 450 nm. 

### 4.11. In Vivo Study

#### 4.11.1. Ethical Statement

The German local authority approved all animal experiments: Landesamt für Landwirtschaft, Lebensmittelsicherheit und Fischerei Mecklenburg-Vorpommern (7221.3-1-066/18), under the German animal protection law and the EU Guideline 2010/63/EU. Mice were bred in the animal facility of the University Medical Center in Rostock under specific pathogen-free conditions. All animals received enrichment in the form of mouse-igloos (ANT Tierhaltungsbedarf, Buxtehude, Germany), nesting material (shredded tissue paper, Verbandmittel GmbH, Frankenberg, Deutschland), paper roles (75 × 38 mm, H 0528–151, ssniff-Spezialdiäten GmbH), and wooden sticks (40 × 16 × 10 mm, Abedd, Vienna, Austria). During the experiment, mice were kept in type III cages (Zoonlab GmbH, Castrop-Rauxel, Germany) at 12-h dark:light cycle, the temperature of 21 ± 2 °C, and relative humidity of 60 ± 20% with food (pellets, 10 mm, ssniff-Spezialdiäten GmbH, Soest, Germany) and tap water ad libitum.

#### 4.11.2. Experimental Protocol

Xenografts were generated by injecting 5 × 10^6^ cells of UT-SCC-14 or UT_SCC-15 (in 50 µL PBS) subcutaneously in the right flank of 6–8 weeks old female NMRI Foxn1^nu^ mice. Two weeks later, mice bearing tumors of ~50 mm^3^ were allocated to treatment groups (Figure 7). Tumor diameters were measured with caliper every three to four days. Tumor volumes were calculated as (length × width^2^)/2. Mice were euthanized before tumors reached 1500 mm^3^. Tumors were embedded in Cryomatrix (Thermo Scientific, Darmstadt, Germany) and used for HE staining. 

### 4.12. Statistics

All values are expressed as mean ± SD (in vitro analysis) or mean ± SEM (in vivo therapy approach). Differences between controls and treated cells were determined by using one-way *ANOVA* (Bonferroni’s Multiple Comparison Test) after proving the assumption of normality (Shapiro-Wilk test). If normality failed, the Kruskal Wallis test was applied. This information is given in the figure captions. Kaplan-Meier survival analysis was done by applying the log rank (Mantel Cox) test. Statistical evaluation was performed using GraphPad PRISM software, version 8.0.2 (GraphPad Software, San Diego, CA, USA). The criterion for significance was set to *p* < 0.05.

## 5. Conclusions

Cyclin-dependent kinase inhibitors (CDKi) have broad therapeutic potential. Here, we show that CDKi’s can be combined with standard cytostatic drugs and that dual CDK inhibition is at least as successful as CDKi/drug combinations. These findings contribute to our understanding of how the treatment of HNSCC can be improved prospectively. The complex effects exerted by specific CDKi-combinations include apoptotic and necrotic cell death as well as methuosis, an uncommon form of cell death, associated with vacuolization of macropinosome and endosome compartments. Dinaciclib and THZ1 were most effective and even better in combination with 5-FU. Another novel finding is the impact on actin fibers and motility properties of tumor cells by specific CDKi’s. Prospective studies should focus on the effects on immune cells—especially because of the CDKi’s potential to increase tumor immunogenicity. 

## Figures and Tables

**Figure 1 cancers-13-02396-f001:**
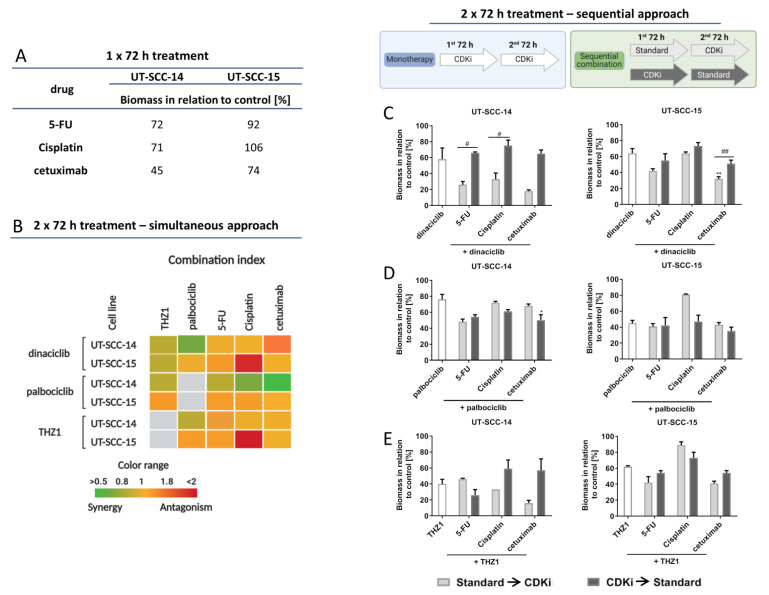
Simultaneous and sequential treatment schedules. (**A**) Biomass quantification after monotherapy with 5-FU, Cisplatin, and cetuximab (1 × 72 h). Doses used here were determined before using classical dose response curve analysis. Read out was done by crystal violet staining and biomass in relation to untreated controls quantified. In (**B**) the Bliss Independence model was used to calculate potential synergistic effects. The green color indicates a synergistic and red color an antagonistic effect of the simultaneous combinations. (**C**–**E**) Sequential treatment: (**C**) dinaciclib [0.005 µM], (**D**) palbociclib [1 µM], and (**E**) THZ1 [UT-SCC14: 0.02 µM; UT-SCC-15: 0.005 µM] in comparison to 2 × 72 h CDKi monotherapy (first bar of each graph). Drug doses were as follows: 5-FU [0.32 µg/mL]; Cisplatin [UT-SCC14: 0.5 µg/mL; UT-SCC-15: 0.05 µg/mL]; cetuximab [0.5 µg/mL]. Mann Whitney U-test (*n* = 3–4 independent experiments) # *p* < 0.05, ## *p* < 0.01 vs. 1st CDKi; Kruskal Wallis test (*n* = 3–4 independent experiments) * *p* < 0.05, ** *p* < 0.01 vs. monotherapy. The 1 × 72 h monotherapy with 5-FU, Cisplatin or Cetuximab confirms that the potential enhancing effect of sequential combination therapy is not due to the single administration of these substances but the effect of the 2 × 72 h CDKi monotherapy (in the left bar) is actually enhanced.

**Figure 2 cancers-13-02396-f002:**
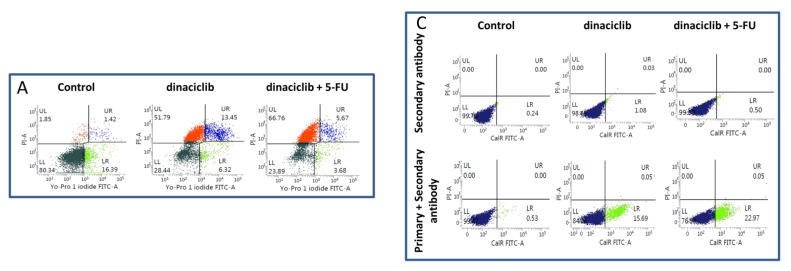
Apoptosis/necrosis assay and detection of immunogenic cell death (ICD). For (**A**,**B**) apoptosis/necrosis assay, cells were stained with Yo-Pro 1 iodide and PI. Cells that were positive for Yo-Pro 1 iodide were defined as early apoptotic, cells that were positive for PI were defined as necrotic, and double positive cells were defined as late apoptotic. Apoptosis/necrosis assay was done after 24 h and 72 h. (**A**) Representative dots plots showing distribution of viable and dead cells (either apoptotic or necrotic). (**B**) Quantitative analysis of apoptotic and necrotic cells subdivided into early apoptotic (Yo-Pro1^+^/PI^−^), late apoptotic (Yo-Pro1^+^/PI^+^) and necrotic ((Yo-Pro1^−^/PI^+^). (**C**,**D**)) ICD was detected after 72 h treatment by staining CalR on the cell surface. In both assays, 10,000 events were measured and the percentage of cells showing CalR translocation and the mean fluorescence intensity (MFI) of CalR^+^ cells are provided. Drug doses were as follows: dinaciclib [0.02 µM]; palbociclib [1 µM]; THZ1 [UT-SCC14: 0.02 µM; UT-SCC-15: 0.005 µM]; 5-FU [90 µg/mL]; Cisplatin [0.1 µg/mL]. (**B**) Kruskal Wallis Test (*n* = 4–5 independent experiments); late apoptotic # *p* < 0.05 vs. control; necrotic * *p* < 0.05, ** *p* < 0.01 vs. control (**D**) 1way ANOVA (*n* = 3–4 independent experiments) * *p* < 0.05, ** *p* < 0.01, *** *p* < 0.001 vs. control.

**Figure 3 cancers-13-02396-f003:**
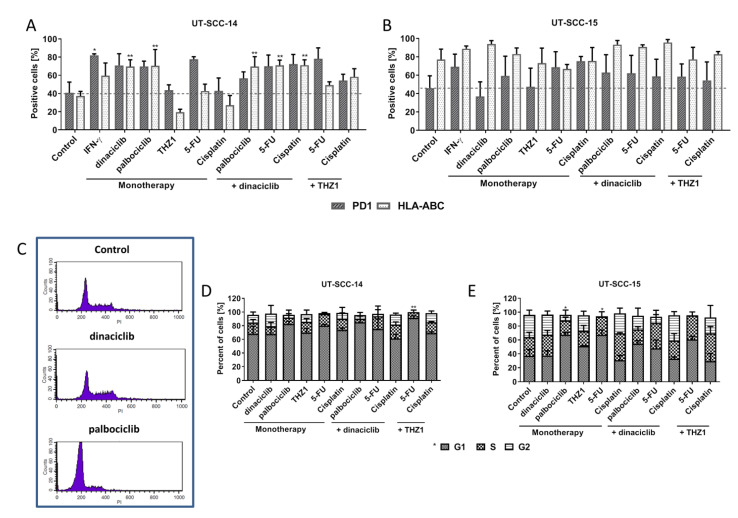
Phenotyping and cell cycle analysis. Phenotyping of (**A**) UT-SCC-14 and (**B**) UT-SCC-15 cells using multi-color flow cytometry. Therefore, cells were stained after 48 h treatment with test substances using the following antibodies: anti-HLA-ABC antibody (MHC I) and anti-CD279 (PD-1). Drug doses were as follows: IFN-γ [50 ng/mL]; dinaciclib [0.02 µM]; palbociclib [1 µM]; THZ1 [UT-SCC14: 0.02 µM; UT-SCC-15: 0.005 µM]; 5-FU [90 µg/mL]; Cisplatin [0.1 µg/mL]. 1 way ANOVA (*n* ≥ 3 independent experiments) * *p* < 0.05, ** *p* < 0.01, vs. control. (**C**–**E**) Cell cycle analysis. Ethanol-fixed cells were stained with PI. (**C**) Representative histograms showing distribution of cell cycle phases in control cells and upon therapy. (**D**,**E**) Quantitative cell cycle analysis showing amounts of cells in G1, S, and G2 phase. Drug doses were as follows: dinaciclib [0.005 µM]; palbociclib [1 µM]; THZ1 [UT-SCC14: 0.02 µM; UT-SCC-15: 0.005 µM]; 5-FU [90 µg/mL]; Cisplatin [0.1 µg/mL]. 1way ANOVA (*n* = 3 independent experiments) * *p* < 0.05, ** *p* < 0.01 vs. control.

**Figure 4 cancers-13-02396-f004:**
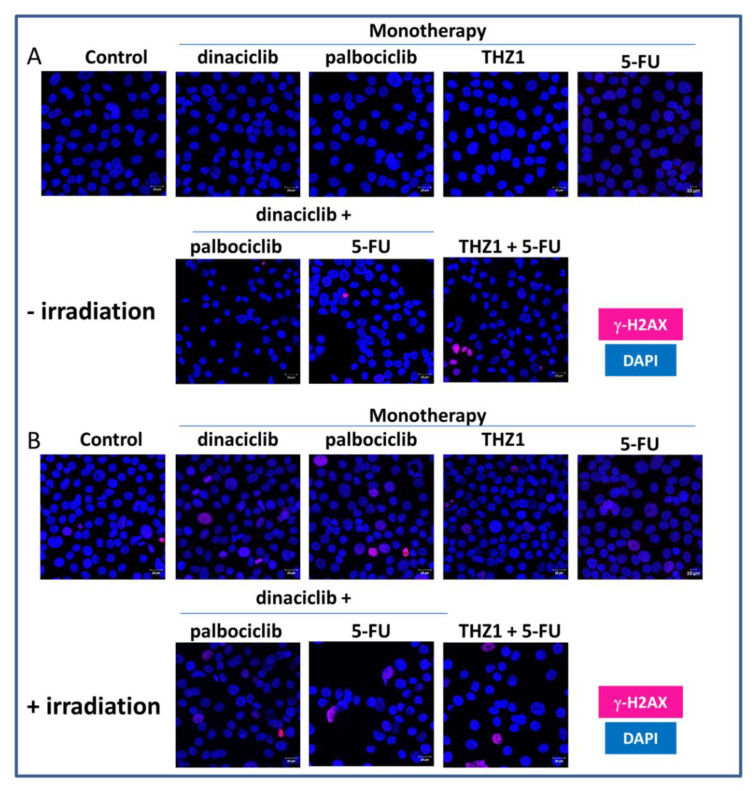
γ-H2AX staining of UT-SCC-14 cells. In order to detect a potential radiosensitizing effect of the test substances, cells were treated 24 h with selected monotherapy and combination therapies and then irradiated with 2 Gy using an IBL637. (**A**,**B**) We had three control groups. The first was completely untreated, the second was treated with the test substances but not irradiated, and the third was only irradiated but not treated with the test substances. Drug doses were as follows: dinaciclib [0.005 µM]; palbociclib [1 µM]; THZ1 [0.02 µM]; 5-FU [0.32 µg/mL]; γ-H2AX staining was performed 6 h after irradiation. Cell nuclei were stained with DAPI. Images were taken on a Zeiss LSM-780 Confocal Laser Microscope.

**Figure 5 cancers-13-02396-f005:**
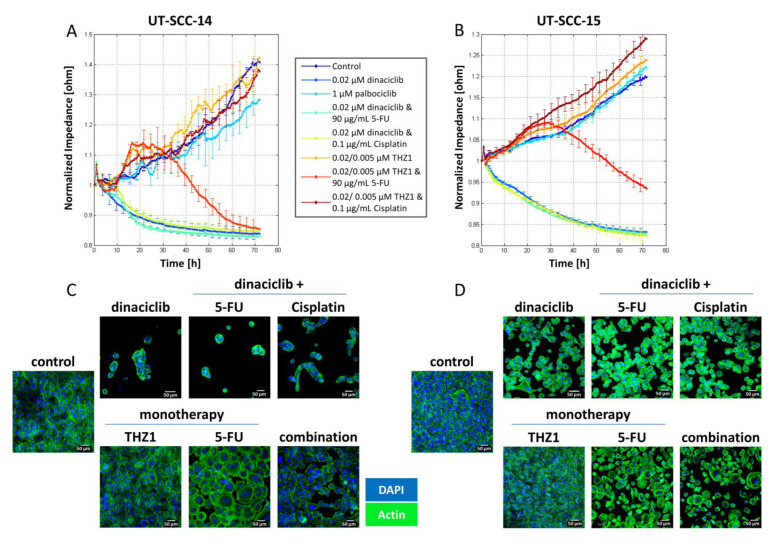
Impedance measurement and cytoskeletal staining. (**A**,**C**) UT-SCC-14 and (**B**,**D**) UT-SCC-15 cells. Cells were seeded in a 96-well ECIS array plate with 20 interdigitated electrodes/well and treated with selected test substances to investigate the impact of the treatment schedules. Drug doses were as follows: dinaciclib [0.02 µM]; palbociclib [1 µM]; THZ1 [UT-SCC14: 0.02 µM; UT-SCC-15: 0.005 µM]; 5-FU [90 µg/mL]; Cisplatin [0.1 µg/mL]. Impedance was monitored in real-time. The analysis of cell-cell contacts was performed by 4000 kHz using ECIS Software. Then, actin staining was performed with phalloidin green. Cell nuclei were stained with DAPI. Analysis was performed with a Zeiss LSM-780 Confocal Laser Microscope. Original magnification 200×.

**Figure 6 cancers-13-02396-f006:**
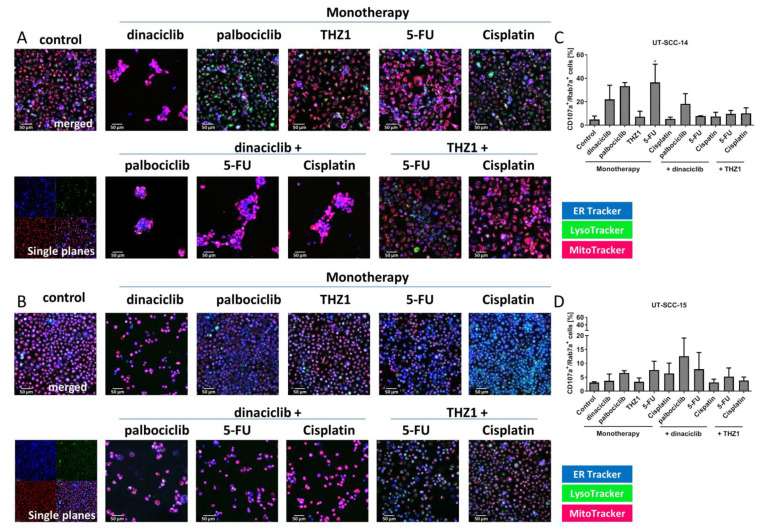
**Influence on mitochondria, lysosomes, ER, and vacuole formation** (**A**,**C**) UT-SCC-14 and (**B**,**D**) UT-SCC-15 cells. (**A**,**C**) To investigate the effect of the test substances on the mitochondrial activity, the lysosome formation, and the ER, cells were treated for 72 h with test substances and stained with MitoTracker (red), LysoTracker (green), and ER-Tracker (blue). Drug doses were as follows: dinaciclib [0.02 µM]; palbociclib [1 µM]; THZ1 [UT-SCC14: 0.02 µM; UT-SCC-15: 0.005 µM]; 5-FU [90 µg/mL]; Cisplatin [0.1 µg/mL]. Representative merged images are shown. For the control, a separated fluorescent image is shown. Analysis was performed with a ZEISS Elyra 7 Confocal Laser Microscope. (**C**,**D**) Cells were stained for CD107a and Rab7a as a hint for vacuole formation and measured via flow cytometry. The percentage numbers of double positive cells are shown. Drug doses were as follows: dinaciclib [0.02 µM]; palbociclib [1 µM]; THZ1 [UT-SCC14: 0.02 µM; UT-SCC-15: 0.005 µM]; 5-FU [90 µg/mL]; Cisplatin [0.1 µg/mL]. 1way ANOVA (*n* = 3 independent experiments) * *p* < 0.05 vs. control.

**Figure 7 cancers-13-02396-f007:**
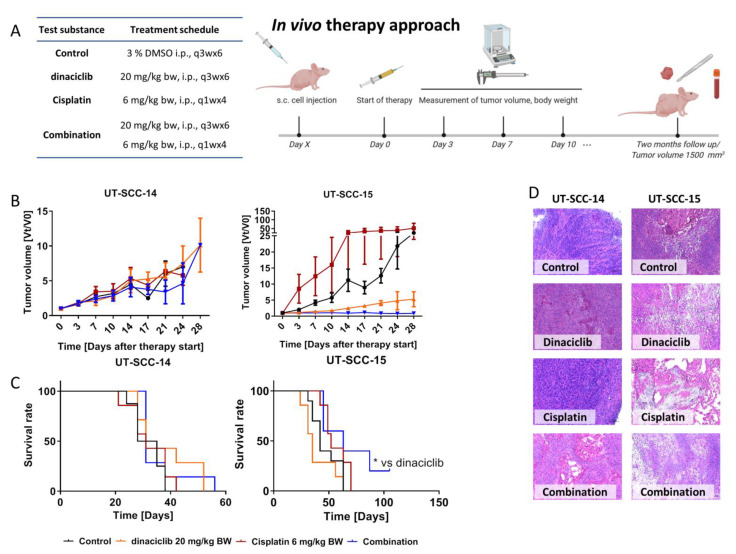
In vivo therapy approach. (**A**) Schematic overview over the treatment protocol with given doses of the test substances. (**B**) Tumor growth curve. Tumor volume was calculated as tumor volume at day × (Vt) divided through the tumor volume at the therapy start (V0). (**C**) Kaplan-Meier survival curve and Log-rank (Mantel-Cox) test. UT-SCC-14: control (*n* = 8 mice); Cisplatin/dinaciclib/combination (*n* = 7 mice/group); UT-SCC-15: control (*n* = 10 mice); Cisplatin/dinaciclib (*n* = 7 mice/group); combination (*n* = 5 mice); * *p* < 0.05 vs. dinaciclib. (**D**) Representative images of the HE stained tumors of each treatment group. Magnification 20×, Scale bar.

## Data Availability

The data presented in this study are available in this article (and Appendix A).

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
