# Peer review of "The Individual Effects of Cyclin-Dependent Kinase Inhibitors on Head and Neck Cancer Cells—A Systematic Analysis"

_cancers, 2021, doi:10.3390/cancers13102396_

Round 1

Reviewer 1 Report

Here, the authors aim to assess the potential for the use of CDK inhibition (CDKi) as single and combination therapies with licensed therapeutics in head and neck cancers. They use a diverse range of in vitro and in vivo approaches to characterise their effects. The use of primary cell lines, increases the significance of the results and the clinical significance. The order of combined therapies is evaluated and reveals some interesting observations with potential clinical applications.

There are currently some issues with the experimentation that would need to be addressed and some minor formatting issues that require attention. I have separated my comments into major, which requires some additional data or controls and minor comments.

Major comments:

  1. In many of the Figures the graph axes are very small and are illegible. In addition, there are figures that have complex data that have been extracted from representative data which are not labelled in Fig2A, B both have 4 additional graphs and Fig 3C, D both have 2 graphs without label. This makes their description and analysis difficult to follow in main text.
  2. General point, in several panels experimental replicates are given as n>5. It would aid transparency to include the exact replicate numbers for each experiment in all figure panels.
  3. Figure 1. I found this figure difficult to interpret as the labels: 1. standard. 1 CDKi on fig 1C-E lacked clarity. A cartoon with treatments and timings would help here.

In addition, I feel that there are some additional insights that could be gained from this approach. The use of CDKi will affect cell cycle stage. This has implications for the approved therapies, that mediate their effects by  nucleotide depletion/DNA replication stress, intrastrand crosslinks or EGFR blockade respectively.

These approaches will have differential effects dependent on cell cycle stage. For example CDK4/6 inhibition would enrich for G1 cells that may prove to be more sensitive to DNA replication stress and nucleotide depletion. Yet, at present, there are no analyses of cell cycle phase after treatment and effects of the inhibitors may therefore be related to cell cycle dynamics. As this could be important for their basis of action, I would recommend this analysis is included.

3. Lack of labelling in Fig 2A, B with the additional graphs lead to a difficult to understand description on results in line 145-159. In addition, the small fonts make this difficult to read. I would suggest either  addition of labels and cross reference in text to improve clarity or replace with a summary table. . There is also a typo in the legend (line 137, Fig1 Iodide PI).

4. On lines 129-134, the authors' state:

"Additionally to the induced cell death, senescence was studied, since this is a common response to CDK inhibition (supplementary Figure 1). These experiments not only confirmed CDKi induced senescence, but also confirmed the therapeutic potential of selected combinations (i.e. dinaciclib + 5-FU, double CDK inhibition).

The data are not clear on the increase in senescence, with only 2 bars showing greater effect than control untreated after CDKi or combination therapies. In addition, error bars are very large for all treated and control cells, suggesting a lack of statistical significance.

5. In figure 4, font sizes on graphs need to be increased. In addition, the flow cytometry data requires addition controls.

The potential reduction in DNA content (Fig 3C lower panel) could indicate a sub-G1 population identified by the the left shift in PI stain. As the authors have already identified affects on apoptotic and necrotic cell death, this analysis should show the cell cycle profile for all combination therapies in appendices as supplementary data as this can affect interpretation.  

Minor point: Fig3C and D have bar charts that are not labelled.

6. Figure 4. This image lacks the replicates to make the conclusions presented. There is no quantitation of the number of H2AX positive nuclei in each condition, there are no experimental replicates and the images as they stand, do not convincingly show differences between treatments. This raises questions as to the size of the effect and its reproducibility. 

7. Figure 5 monitors differences in cell-cell contacts via impedance. This approach relies on cell-cell contacts to alter the conductivity of the cells. Here, there are significant differences in cell density and cell-cell contacts in dual treated cells in C and D upper panel. In addition, there are no untreated controls for the monotreated cells in the lower panels. This reduction n cell cell contacts likely leads to the changes seen here. As such, I think that the conclusions in line 335-337 is overstated. "The strong antitumoral potential of dinaciclib was further confirmed by impedance measurement at a frequency of 4000 Hz, taking cell contacts as one of the most decisive factors [37]." 

8. There are controls missing in Figure 7, the authors use flow cytometry to quantify vacuole formation without showing the data for a representative experiment or fluorescence microscopy image showing that the labelling is specific.

9. In section 2.7, there is no data showing invasiveness and this term should be removed. The data presented shows a scratch assay that shows migration. data should be discussed in these terms.

Minor points:

lines 379-380 "Palbociclib, THZ1, and 5-FU showed the most promising effects in radiotherapy likewise affected the cell cycle". I do not know what point the authors are making here. It requires clarification.

Lines 335-337 "The strong antitumoral potential of dinaciclib was further confirmed by impedance measurement at a frequency of 4000 Hz, taking cell contacts as one of the most decisive factors [37]."

Reviewer 2 Report

This paper has a Dr. Jekkyl and Mr. Hyde character. The authors have spend considerable time and money (I have some issues with no funding being mentioned) to report a barrage of experiments investigating the potential effect of CDKis in two H&N cancer cell lines. They have a beautiful graphical abstract (I am truly envious of that) and an excellent abstract and conclusion section. The materials and methods need some fine combing.

However, this is where it starts to go south. The data presentation is somewhat dull and feels as a simple presentation without any story or aim behind. The discussion section is confusing and looks like a first draft and not of the same level as the abstract. This will have to be rewritten.

I have also several comments on the pdf. A major issue I see is that there is no justification for the choice of dose and time course of treatments and why in most experiments they are not the same. Also, in their attempt to include as many data as possible most figures are of low quality and even trying to review them is painstaking. It would be a shame for all these data to be unpublished, so extensive reviewing of the draft from the authors is suggested.

Author Response

  1. First, the reviewer wished to revise the materials and methods section. We thank the reviewer for this advice and improved the M&M part accordingly. We additionally followed the reviewers’ suggestions made in the pdf version.
  2. Then, she/he wished the discussion to be We totally agree that the discussion was not presented at high quality and carefully revised the complete section. We hope the new version meets all your expectations.
  3. We also addressed all your comments in the pdf file and updated the manuscript accordingly. Comments made in the pdf were answered directly by us in the file.
  4. Again in the pdf file, the reviewer wanted to see raw data of the experiment shown in Figure 6C. These can be found on page 2 of this file.
  5. Another concern raised by the reviewer is the choice of dose and time course of treatments and why in most experiments they are not the same. For the former, we performed detailed dose-response curves in preceding experiments. Based on this data, we decided to go on with doses below the IC50 to check for synergistic effects in the combinations. The time course of treatment was chosen on a basis of our long-term experience with epithelial cells, considering doubling times and to mimic the in vivo Therefore, cells received two treatment cycles of 72 hours. This has been included in the results section (2.1, lines 86-88). For the latter, we used different concentrations between the two cell lines based on the previous dose-response curve analysis. Sensitivity towards individual drugs was quite heterogeneous and thus doses had to be adopted for each cell line. Apart from this, we used the same doses for each experiment. The cell cycle and mitotic index analysis is the only exception. Here, dinaciclib was used in lower doses (0.005 µM vs. 0.02 µM). The rationale for reducing the dose here is the high toxicity of dinaciclib and the resulting low number of residual cells. In order to allow generation of high quality data, we decided to reduce the dose of dinaciclib in the experiment. We hope this is understandable.
  6. Finally, the reviewer claimed the quality of the figures because of the amount of data included in each figure. We revised the figures carefully and put some information in the legends (e.g. drug doses). We additionally removed flow cytometry data for PD-L1 (Figure 3), since we did not measure any differences in this marker. We are confident that this improved data presentation is less strenuous.

Raw data of Figure 6 C

Table 1: Raw data of CD107a+/Rab7a+ UT-SCC-14 cells.

Control

dinaciclib

palbociclib

 5-FU

 Cisplatin

dinacilcib +

palbociclib

dinacilcib +

5-FU

dinaciclib + Cisplatin

THZ1

THZ1+5-FU

THZ1+

Cisplatin

4,1

44,6

36,9

47,9

3,3

16

8,2

14,6

2,1

15,2

19,4

0,1

3,3

35,7

55,7

4,6

34,2

12

3,2

3,1

5

4

10,4

18,1

27,4

5,8

8,3

4,1

12

4,6

16,6

8,7

7,2

Shown are the values from three independent experiments measured by flow cytometry.

Table 2: Statistical analysis (one-way ANOVA, Dunnett's multiple comparisons test). 

Dunnett's multiple comparisons test

Mean Diff,

95,00% CI of diff,

Significant?

Summary

Adjusted P Value

treatment

Control vs. dinaciclib

-17,13

-46,71 to 12,44

No

ns

0,4678

dinaciclib

Control vs. palbociclib

-28,47

-58,04 to 1,109

No

ns

0,0631

palbociclib

Control vs.  5-FU

-31,60

-61,18 to -2,024

Yes

*

0,0324

 5-FU

Control vs.  Cisplatin

-0,5333

-30,11 to 29,04

No

ns

>0,9999

 Cisplatin

Control vs. dinaciclib + palbociclib

-13,23

-42,81 to 16,34

No

ns

0,7341

dinaciclib + palbociclib

Control vs. dinaciclib + 5-FU

-5,867

-35,44 to 23,71

No

ns

0,9965

dinaciclib + 5-FU

Control vs. dinaciclib + Cisplatin

-2,600

-32,18 to 26,98

No

ns

0,9996

dinaciclib + Cisplatin

Control vs. THZ1

-2,400

-31,98 to 27,18

No

ns

0,9997

THZ1

Control vs. THZ1 + 5-FU

-4,767

-34,34 to 24,81

No

ns

0,9993

THZ1 + 5-FU

Control vs. THZ1 + Cisplatin

-5,333

-34,91 to 24,24

No

ns

0,9970

THZ1 + Cisplatin

We are very confident that the revised version now matches the requirements for publication in Cancers. We would be very pleased if you would find this enhanced version suitable for publication.

In the name of all authors,

Yours sincerely,

Nina Irmscher

Reviewer 3 Report

The authors present a work-up of CDK4/6 inhibitors in HNSCC models, and show that they are effective at inducing phenotypes consistent with other cancer cell line/xenograft models. The novelty of the work is the extension to HNSCC models. Experiments are appear to be technically well done and analyzed. As CDK4/6 inhibitors are already being evaluated clinically in HNSCC (and did not have great effect in recurrent disease), I'm unsure of why the authors went into this depth of analysis on cell line models. For this reason, the impact of this paper will be very low on the field. 

Round 2

Reviewer 1 Report

The authors have addressed my comments and improve the legibility of the figures. The content is more accessible and better presented. I am satisfied with the authors responses and improvements to figures and additional supporting data. Overall, I am now of the view that the paper should be accepted.

Reviewer 2 Report

A significantly better paper than the initial version (p<0.05). :-)

Comments to writers' replies are in the pdf file. I believe the paper should be published in its current form.

Best of luck in future ventures. 
